# Exploration of the Magnetic Flux Density on the Surface of Seamless Knitted Fabrics Manufactured with Magnetic Polypropylene Fibers

**DOI:** 10.3390/ma15030880

**Published:** 2022-01-24

**Authors:** Yimin Xiang, Miao Su, Zimin Jin, Kunying Chen, Jianwei Tao, Zhansong Shi

**Affiliations:** 1College of Textile Science and Engineering, Zhejiang Sci-Tech University, Hangzhou 310018, China; yiminaaa1@163.com (Y.X.); Kivenjin@163.com (Z.J.); cky9128378892021@163.com (K.C.); 2Zhejiang Bangjie Holding Group Co., Ltd., Yiwu 322000, China; taojianweibj@163.com; 3Zhejiang Yunshan Textile Printing Dying Co., Ltd., Lanxi 321100, China; shi8863007@126.com.cn

**Keywords:** magnetic fiber, magnetic flux density, seamless knitting, fabric property

## Abstract

In this paper, magnetic fibers were integrated with seamless knitting technology. Additionally, the raw materials for the outer fabrics and the relevant yarn feed ratio were designed, including the polypropylene yarn with different magnetic powder contents (0%, 10% and 50%) and its yarn feed ratio (100:0, 75:25, 50:50 and 25:75) to graphene viscose yarn. In addition, weft plain stitch, 1 + 1 mock rib and 1 + 3 mock rib were adopted to weave polyamide fiber/polyurethane fiber wrap yarn as the lining materials into 12 knitted fabric samples on a seamless knitting machine according to the partial addition method in the orthogonal experimental design. As per the test and analysis results of the magnetic flux density on the front and back surfaces of 12 seamless knitted fabrics, polypropylene yarn with different magnetic powder contents in outer fabrics is the most significant factor affecting the magnetic flux density on the surface, followed by the yarn feed ratio of outer fabrics and fabric stitches. The findings in this study can provide a reference and theoretical basis for the specification design of seamless knitted fabrics manufactured by magnetic fabrics to a certain extent.

## 1. Introduction

In contemporary society, due to the accelerated pace of life, the working style of prolonged sitting, inappropriate posture and increasingly delayed sleep from over-reliance on electronic products, most people are in a condition of sub-health, in which the vitality, various functions and adaptations of the body decline to varying degrees over a certain period of time, and potential disease cannot be detected without thorough systematic medical examination [1]. According to statistics, there is a growing trend towards a younger age [2].

Some magnetic therapy products based on permanent magnets can be locally applied to specific parts of the human body, with a strong ability to perform penetration and time control and usually characterized by low cost and favorable safety. Therefore, a certain number of people in the world are using magnetic therapy products based on permanent magnets to improve some minor diseases or sub-health conditions. In recent years, it has been demonstrated in some scientific experimental facts that static magnetic fields have certain effects in regulating blood flow, improving the skeletal system, relieving pains, affecting metabolism, relieving inflammation and promoting wound healing [3]. Additionally, magnetic fields have analgesic, sedative, spasmolytic, anti-inflammatory, discutient and antihypertensive effects, and they can also be employed to treat diseases, strengthen physique and prolong life [4]. In terms of analgesia, magnetic fields can be adopted to relieve neuropathic pains and rheumatic pains, and they are also effective for the mitigation of spastic pains and radiating pains [5]. For example, constant magnetic fields have been applied for the treatment of dysmenorrhea [6], fibromyalgia [7], knee pains and chronic low back pains [8], in which remarkable curative effects have been achieved. The exploration of the magnetic flux density on the surface of seamless knitted fabrics manufactured with magnetic polypropylene fibers contributes to promoting the development of this kind of clothing fabric, thus, effectively improving the sub-health condition of the human body.

Currently, with the rapid advancement of seamless knitting technology, seamless knitted garments can be integrally formed with only a small amount of stitched and spliced fabrics, and knitted underwear with high-elasticity and close-fitting properties can be directly manufactured, which combines fit, comfort and fashion. In this paper, combined with modern people’s demand for clothing with health care functions, the magnetic polypropylene fibers (MPFs) with different magnetic powder contents were applied to the seamless knitting field. Further, the effects of different raw materials for outer fabrics and relevant yarn feed ratios and different stitches on the magnetic flux density on the surface of magnetic seamless knitted fabrics were explored. The findings of this study provide a theoretical basis for the application of magnetic health care textiles in physical therapy of chronic diseases, such as pains in the shoulders, neck, waist and legs and also provide reference for the design and manufacture of magnetic health care clothing. It has considerable practical significance and broad development prospects [9] and is a practical study that can be applied to medical textiles.

## 2. Establishment of Sample Schemes for Seamless Knitted Fabrics

### 2.1. Determination of Yarn Selection Schemes

Considering that magnetic strength is directly related to magnetic powder content, polypropylene yarn was purchased directly from Beijing Baiquan Chemical Fiber Factory (Beijing, China). The magnetic powder was strontium ferrite (0.5–2 μ) in permanent ferrite material [10] and then magnetic polypropylene fiber was prepared by the blending spin method. Finally, the ordinary polypropylene yarn with 0% magnetic powder content (MPF-0), magnetic polypropylene yarn with 10% magnetic powder content (MPF-10) and magnetic polypropylene yarn with 50% magnetic powder content (MPF-50) were used. Considering the poor moisture absorption of the polypropylene fibers, the production cost and the heat retention, as well as the antistatic behavior of winter clothing, graphene viscose yarn was selected. Meanwhile, the body-fitting and body-shaping yarn was selected for close-fitting, seamless knitted fabrics, and polyamide fiber/polyurethane fiber wrap yarn with favorable elasticity was generally selected as lining raw materials. MP yarn and graphene viscose yarn were used as raw materials for the face yarn, and the nylon/spandex coated yarn (22.2dtex(20D) nylon yarn/77.8dtex(70D) spandex yarn) was used as raw material for the lining yarn. 

In order to avoid the influence of manufacture, weaving and other processes on equipment, the magnetic fibers on the market are not magnetized, namely, they are not magnetic. Instead, magnetic fibers are woven into magnetic textile fabrics or magnetic clothing and then magnetized to make permanent magnetic nanoparticles in magnetic fibers generate magnetic fields. In this paper, magnetic fabrics were magnetized, and a magnetizing machine was employed to perform the magnetization, which quickly realized the saturated magnetization for the fabrics [11]. At room temperature, magnetic fields were applied to the magnetic polypropylene fibers, and the permanent magnetic nanoparticles in the magnetic fibers were magnetized instantly, thus, forming a constant magnetic field on the surface of the magnetic fabrics. Before the permanent magnetic nanoparticles reached magnetic saturation, the surface magnetic flux density increased with the increase of the applied magnetic field intensity [12]. Specific specifications for the seamless knitting yarn are listed in Table 1.

### 2.2. Determination of Stitches

(1)Weft Plain Stitch

The plain stitch is one of the basic stitches of single-sided, weft-knitted fabric. Knitted fabrics with weft plain stitches are characterized by favorable extensibility in longitudinal and transverse directions, as well as a smooth front surface of the fabrics. 

(2)1 + 1 and 1 + 3 Mock Rib Stitches

The back surface of the knitted fabrics with mock rib stitches is floating thread, and the front surface has the appearance of rib stitches, which contributes to the thickness and heat retention properties of the fabrics. 

Considering that the focus of this study was mainly placed on the seamless knitted fabrics manufactured with magnetic polypropylene fibers used in winter, the most common weft plain stitch, 1 + 1 mock rib and 1 + 3 mock rib were selected in this paper, combined with the use of these products, with the stitches shown in Figure 1.

The properties of knitted fabrics are, in general, related to their structure [13]. As can be seen from Figure 1, the stitches of 1 + 3 mock rib are flat and thick due to the floating threads on the back surface, those of 1 + 1 mock rib are flat and durable due to the floating threads on the back surface and the weft plain stitches are flat and smooth. 

### 2.3. Determination of the Yarn Feed Ratio of Outer Fabrics

In this paper, the heat retention and antistatic properties of winter clothing were achieved by changing the yarn feed ratio of polypropylene yarn with different magnetic powder contents to graphene viscose yarn in the outer fabrics. The yarn feed ratio of polypropylene yarn with different magnetic powder contents and graphene viscose yarn in outer fabrics was reasonably set as follows: 8 feeders of polypropylene yarn with different magnetic powder contents + 0 feeders of graphene viscose yarn, with the ratio of 100:0; 6 feeders of polypropylene yarn with different magnetic powder contents + 2 feeders of graphene viscose yarn, with the ratio of 75:25; 4 feeders of polypropylene yarn with different magnetic powder contents + 4 feeders of graphene viscose yarn, with the ratio of 50:50; 2 feeders of polypropylene yarn with different magnetic powder contents + 6 feeders of graphene viscose yarn, with the ratio of 25:75.

### 2.4. Determination of the Yarn Feed Ratio of Outer Fabrics

In this paper, an exploration was mainly performed on the influence of polypropylene yarn with different magnetic powder contents, its yarn feed ratio to graphene viscose yarn and fabric stitches on the properties of magnetic fabrics. In addition, the sample scheme was designed by the neat comparison and balanced disperser in the orthogonal experimental design [14]. The specific factor levels are listed in Table 2.

In Factor A, the yarn feed ratio of outer fabrics included four levels; in Factor B and Factor C, the polypropylene yarn with different magnetic powder contents in outer fabrics and fabric stitches included three levels, respectively. Therefore, it was a multi-factor and multi-level experiment. In this paper, the partial addition method in the orthogonal experimental design [15] was selected to design the L9 (34) +3 orthogonal experiment. The additional level of the yarn feed ratio of outer fabrics in Factor A was 25:75. Specifications of fabric samples are listed in Table 3.

## 3. Exploration of the Magnetic Flux Density on the Surface of Fabrics

Magnetic fabrics belong to the category of functional and wearable fabrics, and those magnetic fabrics with magnetic health care function possess great market potential. The research and development of this kind of health care fabric have become hot spots in the textile and garment industry at present [16]. The surface magnetic flux density of 12 seamless knitted fabrics was tested in an attempt to explore the influence of different raw materials used for manufacturing outer fabrics, relevant yarn feed ratio and different stitches on the surface magnetic flux density of these fabrics. 

The magnetic field on the surface of magnetic fabrics belongs to the category of the low-intensity magnetic field. In this paper, the magnetic field on the surface of magnetic fabrics was characterized by magnetic flux density *B*. When the direction of charge motion was perpendicular to that of the zero magnetic force, it received the maximum magnetic force (*F*_max_) [9], and the magnetic flux density *B* at this point could be obtained by Equation (1): (1)$$B=\frac{F_{\max}}{qv}$$
where *q* represents the electric quantity of moving charge; *v* represents the rate of moving charge; *F*_max_ represents the maximum magnetic force on the moving charge; and B represents the magnetic flux density, in Tesla (Tex)/Gauss (Gs), with the conversion relationship between both being 1 T = 10^4^ Gs.

### 3.1. Test Principle of Magnetic Flux Density on the Surface of Fabrics

(1)Test Instrument and Test Principle

Currently, the Tesla/Gauss meter based on the Hall effect principle is mainly adopted to test the magnetic flux density on the surface of magnetic fabrics. In this instrument, the probe of the Hall element is adopted to detect the surface magnetic flux density on the surface of magnetic fabrics with high resolution, high precision and sensitive response. When the Hall element in the probe is employed to measure the magnetic flux density on the surface of fabrics according to the Hall effect principle, the Hall voltage *U_H_* can be obtained, as expressed in Equation (2).
(2)$$U_{H}=K_{H}I_{H}B\cos\theta$$
where *K_H_* represents the Hall coefficient, *I_H_* represents the working current, *B* represents the magnetic flux density and *θ* represents the included angle between the Hall element plane and the normal of the magnetic field direction [9].

When *θ* = 0, namely, when the Hall element plane is perpendicular to the direction of the magnetic field, UH can reach the largest value, UH is proportional to the measured magnetic field and the measured magnetic flux density B can reach the largest value, correspondingly [9]. Due to the fact that the magnetic powder in magnetic fibers is permanent magnetic nanoparticles, each permanent magnet nanoparticle is equivalent to a magnet with a N pole and S pole, which can emit magnetic induction lines, thus, forming a closed loop from N pole to S pole. Additionally, the magnetic induction lines are curves or straight lines, which do not intersect or become turned. Due to the uniform distribution of permanent magnetic nanoparticles in magnetic fibers and on the surface of magnetic fabrics, the magnetic field on the surface of magnetic fabrics is also uniform, with the maximum and minimum values of the N pole and S pole. After wearing magnetic fabrics, the fabric covers the surface of the human body to form a curved surface with radians, and the magnetic induction lines in different directions contact with the human skin. For the reason that the magnetic field acting on the human body is the normal magnetic field in the vertical direction, the normal magnetic flux density of magnetic field on the surface of fabrics was explored in this paper. The test results of the magnetic flux density on the surface of fabrics are shown in Figure 2, and the distance δ between the Hall element plane and probe end face was 0.5 mm ± 0.05 mm.

The magnetic field on the surface of magnetic fabrics belongs to the category of low-intensity magnetic field, which requires that Tesla/Gauss meter used for the testing of magnetic fabrics has a resolution of not less than 0.001 mT and a relative measurement mode so that the instrument can perform the zero calibration in the test environment, respond sensitively to the subtle changes of magnetic field and ensure that the value measured by the instrument is consistent with or as close as possible to the true value of the magnetic field on the surface of fabrics as possible. The instrument selected in this paper was a CH-1600 all-digital Gauss/Tesla meter, with a resolution of 1 × 10^−4^ mT.

(2)Test Environment

Due to the fact that the surface magnetic field of magnetic fabric belongs to the low-intensity magnetic field, it is necessary to be rid of the influence from the test environment as much as possible, in an attempt to improve the precision and accuracy of the measured value. The Hall probe is a high-precision, vulnerable item, and the Hall element is a sensor made of high-precision semiconductor materials which is not only a magnetic sensitive material but also a temperature sensitive material [17]. It is necessary to perform the zero calibration for the Hall probe before testing to eliminate the influence of probe zero drift or micro-magnetic field in the test environment. The drift value of the Tesla meter based on the Hall effect is generally below 0.01 mt. The experiment shows that the drift of the instrument does not change significantly with the change of temperature in the temperature range of 0 °C to 40 °C, and the influence on the measurement results of magnetic induction intensity on the fabric surface is very weak and can be ignored. Therefore, the measurement of fabric surface magnetic field with Tesla meter can be carried out in ordinary room temperature environment. Therefore, the humidity in the test environment was kept constant, and the test temperature was approximately 20 °C, which conduces to avoiding the influence of humidity and temperature on the test value as much as possible. Meanwhile, the test was conducted free from strong electromagnetic field interference, and the instrument was not placed under the air outlet or air conditioner.

### 3.2. Test of Magnetic Flux Density on the Surface of Fabrics

Experimental Instrument: CH-1600 all-digital Tesla/Gauss meter, as shown in Figure 3.

Experimental Procedure: Refer to the standard FZ/T01116-2012 in the textile industry. Firstly, the samples were placed in the environment with a constant temperature and humidity for humidity adjustment for 24 h; the axial end face of the probe was vertically contacted or flush with the surface of the test bench for zero calibration; and the end face of the Hall probe was contacted with the test surface of the fabric samples laid flat on the test bench. Subsequently, the magnetic flux density on the surface of the fabrics was tested. During the test, the probe was subjected to relative motion. Five samples 15 cm × 15 cm in size were cut for each fabric, and the test results were the average surface magnetic flux density of the five specimens. In this paper, the center point with the size of 1.5 cm × 1.5 cm was selected as the test point, which was divided into ten rows and ten columns along the transverse and longitudinal directions. Each row and column was distributed with ten test points. The sample with the size of 15 cm × 15 cm was evenly divided into 100 test points, with the specific distribution shown in Figure 3.

### 3.3. Test Result Analysis of Magnetic Flux Density on the Surface of Fabrics

Twelve seamless knitted fabrics were subjected to the surface magnetic flux density test, with the test results listed in Table 4.

As per the test result analysis from Table 4, the surface magnetic flux density of 12 seamless knitted fabrics can be ranked as #3 > #6 > #9 > #12 > #5 > #2 > #8 > #11 > #10 > #7 > #4 > #1. The surface magnetic flux density of the MPF-50 fabric was the largest, followed by that of the MPF-10 fabric, and the MPF-0 fabric had the smallest surface magnetic flux density. In terms of the average magnetic flux density, the maximum magnetic flux density of the N pole and S pole on the surfaces of fabrics #3, #6, #9 and #12 had the largest values and #2, #5, #8 and #11 had relatively small values, while #1, #4, #7 and #10 had the smallest values.

In this paper, the relationship between the surface magnetic flux density and the stitch parameters of 12 seamless knitted fabrics was explored by orthogonal experiments and intuitive analysis [18], with specific results listed in Table 5.

According to the orthogonal analysis results of the surface magnetic flux density of 12 seamless knitted fabrics obtained in Table 5: (1) Direct observation: when the surface the magnetic flux density of the fabric was the highest (0.5571), the corresponding levels of A, B and C were 1, 3 and 3, respectively, i.e., scheme A1, B3 and C3; (2) Calculation and analysis: firstly, the sum of the index values of the three factors A, B and C at three levels was calculated and recorded as K1, K2 and K3. Then, their arithmetic mean values were calculated as K1j, K2j, K3j and K4j, respectively, and, finally, their range as R, respectively. Through calculation, it can be seen that the level corresponding to the maximum average value of factor A was 1; the level corresponding to the maximum average value of factor B was 3; and the level number corresponding to the maximum average value of factor C was 3, so the best level collocation of the three factors was A1, B3 and C3. By comparing the range R value, we know that the most important factor affecting the magnetic flux density on the surface of fabrics was polypropylene yarn with different magnetic powder contents in outer fabrics (Factor B), followed by the yarn feed ratio of outer fabrics (Factor A) and, finally, fabric stitches (Factor C). Therefore, through the above method, we only conducted 12 tests, selected the better sample scheme, saved the test cost and calculated the order of factors affecting the test results.

In order to draw the test analysis results more visually, we used the method of drawing a trend chart (effect curve) to draw the correct comprehensive analysis conclusion, as shown in Figure 4.

We can see more intuitively from the figure: (1) Under the influence of different yarn feed ratios of outer fabrics (Factor A), K1 > K2 > K3 > K4 can be obtained, namely, when the yarn feed ratio of outer fabrics is 100:0, the magnetic flux density on the surface of fabrics is relatively larger. The higher the content of magnetic fibers in magnetic fabrics, the higher the magnetic flux density of magnetic fabrics; (2) Under the influence of different magnetic powder contents in outer fabrics (Factor B), K3 > K2 > K1 can be obtained, namely, when the raw material used for manufacturing outer fabrics is MPF-50, the magnetic flux density on the surface of fabrics is relatively larger. The magnetic powder content is an important factor affecting the magnetic flux density on the surface of fabrics, and it directly affects the magnetism of magnetic fabrics. The higher the content of magnetic powder in magnetic fibers, the better the magnetic properties of magnetic fibers; (3) Under the influence of different stitches (Factor C), K3 > K1 > K2 can be obtained, namely, when the stitch is 1 + 3 mock rib, the magnetic flux density on the surface of fabrics is relatively larger.

### 3.4. Washing Fastness Test on the Magnetic Flux Density of Fabrics 

After magnetic nanoparticles were added to the polymer solution, magnetic fibers could be obtained through spinning. The magnetic flux density of magnetic fabrics is influenced by the type of magnetic powder in magnetic fibers, the content of magnetic powder and the content of magnetic fibers. In this paper, the magnetic fiber yarn was added with permanent magnetic nanoparticles. The magnetism of magnetic fabrics can be maintained persistently after the treatment of permanent magnetization. In consideration of the influence of the environmental magnetic field, biomagnetic field and wearing and washing on the magnetic flux density of magnetic fabrics, the washing fastness test was conducted on the magnetic flux density of magnetic fabrics in order to investigate the persistency and durability of magnetic seamless knitted fabrics, thus, ensuring the persistent health preservation effect of magnetic fabrics during their application.

In this paper, the washing fastness test on the magnetic flux density of magnetic fabrics was conducted according to GB/T 8629-2017. Due to the fact that the test of the magnetic flux density on the surface of fabrics is susceptible to changes in environmental temperature and humidity, electromagnetic field and other factors, there may be contingency and randomness in the test data. The total average magnetic flux density on the surface of fabrics with MPF-10 outer fabrics is 0.0412–0.0562 mT. The smaller the magnetic flux density on the surface of fabrics, the more susceptible the test is to changes in environmental temperature and humidity, electromagnetic field and other factors. Therefore, four fabrics manufactured with magnetic polypropylene fibers and MPF-50 outer fabrics (#3, #6, #9 and #12) were selected for the washing fastness test on their magnetic flux density. Five samples (each with the size of 15 cm × 15 cm) were selected from each fabric. The A-type standard washing machine was utilized, and cotton-woven fabrics were selected as the co-washed fabrics. The washing times were 5, 10, 15 and 20, respectively. Subsequently, these fabrics were flattened and dried.

After washing and drying treatment, four fabrics manufactured with magnetic polypropylene fibers (#3, #6, #9 and #12) were subjected to the surface magnetic flux density test, which was performed based on the surface magnetic flux density test of the fabrics mentioned above. In addition, the influence of washing times on the changes in the magnetic flux density on the surface of the four fabrics manufactured with magnetic polypropylene fibers (#3, #6, #9 and #12) and the distribution of magnetic flux density on their surface were explored and analyzed as per the test results of the magnetic flux density on the surface of fabrics.

### 3.5. Result Analysis of Washing Fastness Test on the Magnetic Flux Density of Fabrics

Four magnetic fabrics (#3, #6, #9 and #12) were subjected to the magnetic flux density test, with the results listed in Table 6.

According to the test results in Table 6, we can see that, with the increase of washing times, the average value of magnetic flux density, the maximum value of the N pole and the maximum value of the S pole of the fabric samples gradually reduced. In order to more intuitively see the change of data, a line graph related to the changes of the magnetic flux density on the surface of the four fabrics manufactured with magnetic polypropylene fibers (#3, #6, #9 and #12) and the washing times was plotted, as shown in Figure 5.

As can be seen from Table 6 and Figure 5, the magnetic flux density on the surface of the four fabrics manufactured with magnetic polypropylene fibers (#3, #6, #9 and #12) presented a decreasing trend with the increase of washing times, but the decrease was small and not significant. It indicates that the adhesive force of the permanent magnetic nanoparticles within the magnetic polypropylene fibers is larger, and they have favorable washing fastness in terms of their magnetism. At the initial stage of washing (namely, the washing times = 5), the changes in the magnetic flux density on the surface of fabrics were more apparent due to the shedding of permanent magnetic nanoparticles in the seamless knitted fabrics manufactured with magnetic polypropylene fibers, and the decrease was larger but not significant. An analysis of the reason is as follows: the magnetic fiber experiences hygroscopic swelling after washing, the stacking density of magnetic nanoparticles changes, the permanent magnet nanoparticles in the magnetic polypropylene fiber seamless knitted fabric are unevenly distributed, the magnetic flux density on the fabric surface is more significant and there are more magnetic nanoparticles in some areas so the magnetic flux density on the fabric surface is enhanced.

## 4. Conclusions

In this paper, the combined method of magnetic fiber yarn and seamless knitting technology was adopted to design and manufacture 12 seamless knitted fabric samples with different raw materials used for manufacturing outer fabrics, relevant yarn feed ratio and different fabric stitches. Additionally, the functional research and analysis of the samples were conducted. In addition, the orthogonal analysis method was adopted to investigate and analyze the influence of polypropylene yarn with different magnetic powder contents, yarn feed ratio of outer fabrics and different fabric stitches on the fabric properties. As can be seen from the above tables and figures: (1) Polypropylene yarn with different magnetic powder contents (Factor B) in outer fabrics has the most significant influence on the magnetic flux density on the surface of fabrics, followed by the yarn feed ratio of outer fabrics (Factor A) and, finally, fabric stitches (Factor C); (2) The higher the magnetic powder content in the magnetic fiber, the better magnetic flux density of the magnetic fiber; when the yarn feed ratio is 100:0, the surface magnetic flux density of the fabric is better. That is, the higher the content of magnetic fiber in the magnetic fabric, the higher the magnetic flux density of the magnetic fabric; when the stitch is 1 + 3 mock rib, the surface magnetic flux density of the fabric is better because the fabric of 1 + 3 mock rib is flatter and thicker than that of 1 + 1 mock rib and weft plain stitch fabric, with obvious concave and convex lines on the surface and more magnetic fiber content. According to the principle of magnetic field superposition and the strong regionality of magnetic flux density on the surface of magnetic fiber fabric, the surface magnetic flux density of the 1 + 3 mock rib fabric is large. However, although the thickness of 1 + 1 mock rib fabric is larger than that of weft flat knitting, its surface magnetic flux density is smaller than that of weft flat knitting. The reason is that the magnetic particle distribution in the magnetic fiber is uneven, the magnetic field on the fabric surface is regional and the ambient temperature and humidity change affected the experimental data; (3) The magnetic nanoparticles of the magnetic fiber are unevenly distributed. The disordered and irregular arrangement of the N pole and S pole of the permanent magnet particles in the magnetic fiber and the interweaving of the fibers make the overlap and decrease of the N pole and S pole of the permanent magnet particles, which makes the magnetic flux density on the surface of the magnetic seamless knitted fabric regional. The magnetic overlap in one area of the fabric surface is enhanced, and the magnetic weakening in another area is reduced. After different washing times, the magnetic fiber experiences hygroscopic expansion and the movement space of permanent magnet nanoparticles in the magnetic polypropylene fiber seamless knitted fabric increases and the fluidity increases, which makes the regionality of magnetic flux density on the surface of most fabrics more significant after washing, and more magnetic nanoparticles are accumulated in some areas so that the surface magnetic flux density of fabrics is enhanced. After washing some fabrics, the distribution of permanent magnet nanoparticles in the magnetic polypropylene fiber seamless knitted fabric is more uniform than before, and the regional magnetic flux density on the fabric surface is reduced.

## Figures and Tables

**Figure 1 materials-15-00880-f001:**
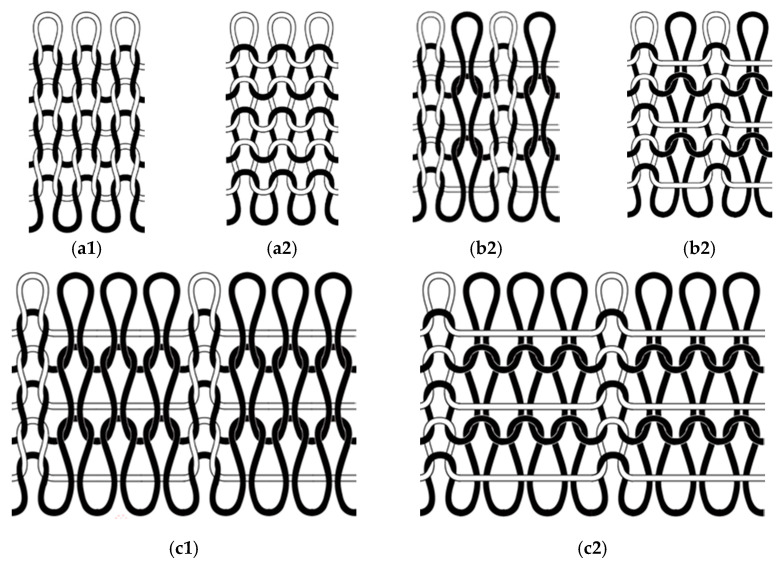
(**a1**) The front surface of weft plain stitches; (**a2**) the back surface of weft plain stitches; (**b1**) the front surface of 1 + 1 mock rib; (**b2**) the back surface of 1 + 1 mock rib; (**c1**) the front surface of 1 + 3 mock rib; (**c2**) the back surface of 1 + 3 mock rib.

**Figure 2 materials-15-00880-f002:**
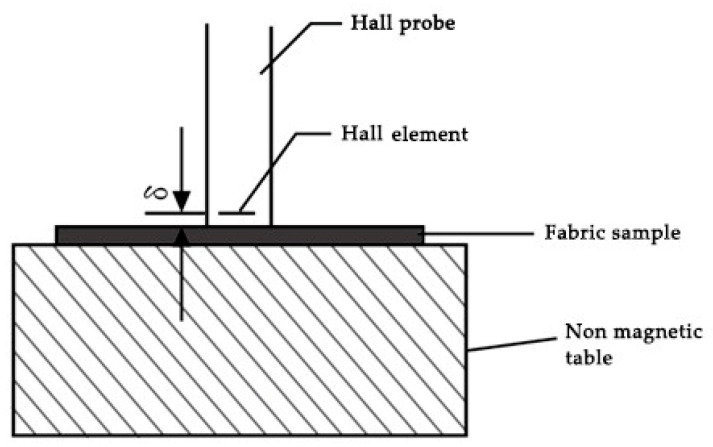
Test diagram of magnetic flux density on the surface of fabrics.

**Figure 3 materials-15-00880-f003:**
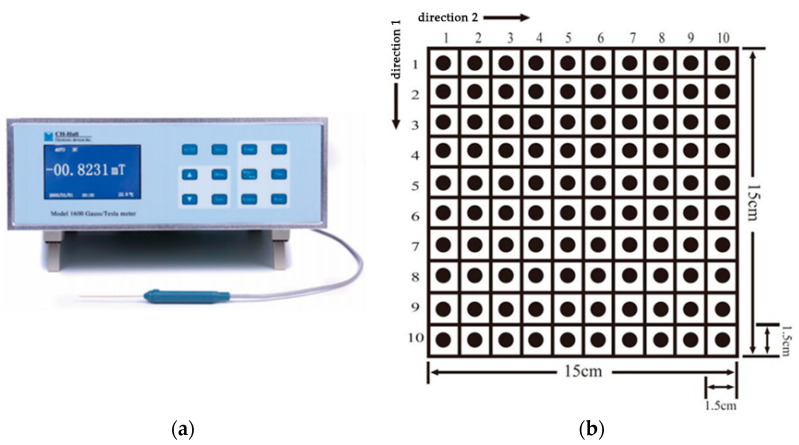
(**a**) CH-1600 all-digital Tesla/Gauss meter; (**b**) test point distribution diagram of the magnetic flux density on the surface of fabrics.

**Figure 4 materials-15-00880-f004:**
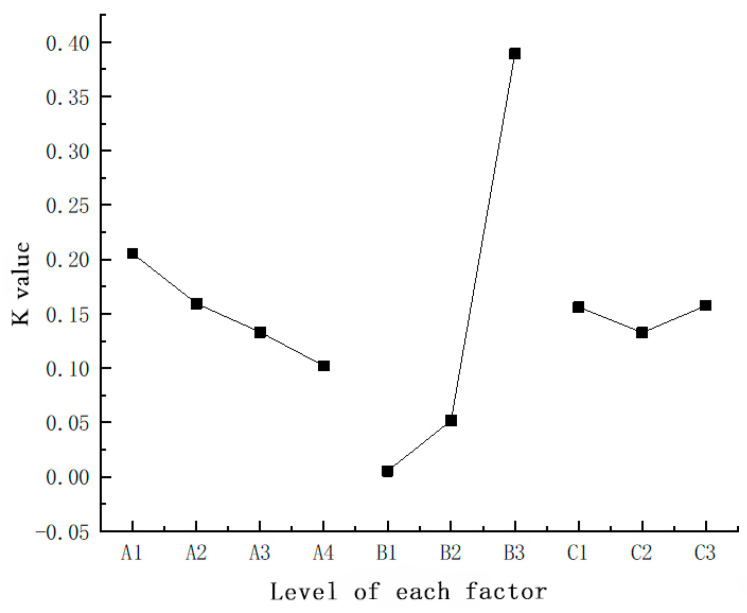
Relationship between the magnetic flux density on the surface of fabrics and fabric stitch parameters.

**Figure 5 materials-15-00880-f005:**
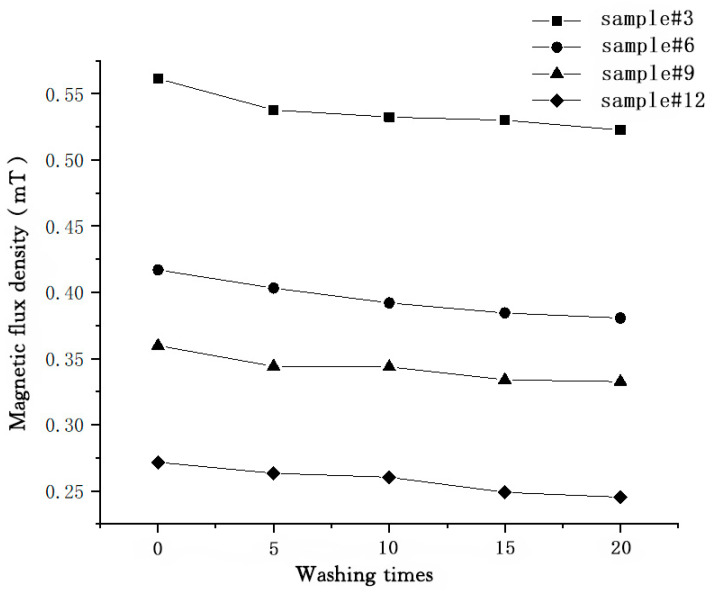
A change graph of the magnetic flux density on the surface of fabrics after different washing times.

**Table 1 materials-15-00880-t001:** Raw materials and specifications of seamless knitting yarn.

No.	Raw Materials for Outer Fabric Yarn	Raw Materials for Lining Yarn
1	122.2dtex(110D) MPF-0	22.2dtex(20D) polyamide fiber/77.8dtex(40D) polyurethane fiber wrap yarn
2	122.2dtex(110D) MPF-10	
3	122.2dtex(110D) MPF-50	
4	106dtex(50s) graphene viscose yarn	

The following: “MP yarn with 50% magnetic powder content (MP-50)” is referred to as “MP-50”; “MP yarn with 10% magnetic powder content (MP-10)” is referred to as “MP-10”; “Common polypropylene yarn with 0% magnetic powder content (MP-0)” is referred to as “MP-0”.

**Table 2 materials-15-00880-t002:** Determination of fabric factor levels.

Factor Level	A Yarn Feed Ratio of Outer Fabrics (Polypropylene Yarn with Different Magnetic Powder Contents: Graphene Viscose Yarn)	B Polypropylene Yarn with Different Magnetic Powder Contents in Outer Fabrics	C Fabric Stitch
1	100:0	MPF-0	Weft plain stitch
2	75:25	MPF-10	1 + 1 mock rib
3	50:50	MPF-50	1 + 3 mock rib
4	25:75	-	-

The “yarn feed ratio of outer fabrics (polypropylene yarn with different magnetic powder contents: graphene viscose yarn)” is hereinafter referred to as “yarn feed ratio of outer fabrics”.

**Table 3 materials-15-00880-t003:** Specifications of fabric samples.

Fabric No.	A Yarn Feed Ratio of Outer Fabrics	B Polypropylene Yarn with Different Magnetic Powder Contents in Outer Fabrics	C Fabric Stitch
#1	100:0	MPF-0	Weft plain stitch
#2	100:0	MPF-10	1 + 1 mock rib
#3	100:0	MPF-50	1 + 3 mock rib
#4	75:25	MPF-0	1 + 1 mock rib
#5	75:25	MPF-10	1 + 3 mock rib
#6	75:25	MPF-50	Weft plain stitch
#7	50:50	MPF-0	1 + 3 mock rib
#8	50:50	MPF-10	Weft plain stitch
#9	50:50	MPF-50	1 + 1 mock rib
#10	25:75	MPF-0	Weft plain stitch
#11	25:75	MPF-10	1 + 1 mock rib
#12	25:75	MPF-50	1 + 3 mock rib

A represents the yarn feed ratio of outer fabrics; A1 is 100:0, A2 is 75:25, A3 is 50:50 and A4 is 25:75. B and C are deduced by analogy.

**Table 4 materials-15-00880-t004:** The surface magnetic flux density of 12 seamless knitted fabrics.

	Test Surface	Front Surface			Back Surface			Total Average Value of the Magnetic Flux Density (mT)
Sample		Average Value (mT)	Maximum Value of N Pole (mT)	Maximum Value of S Pole (mT)	Average Value (mT)	Maximum Value of N Pole (mT)	Maximum Value of S Pole (mT)	
#1		0.0036	0.0133	−0.0092	0.0025	0.0063	−0.0108	0.0031
#2		0.0568	0.1043	−0.1255	0.0557	0.0994	−0.1274	0.0562
#3		0.5614	2.1781	−2.2112	0.5528	2.2064	−1.2310	0.5571
#4		0.0052	0.0195	−0.0056	0.0030	0.0117	−0.0097	0.0041
#5		0.0590	0.1044	−0.1231	0.0589	0.1194	−0.1279	0.0590
#6		0.4169	1.3410	−1.7747	0.4135	1.3372	−1.8742	0.4152
#7		0.0071	0.0001	−0.0172	0.0054	0.0092	−0.0162	0.0063
#8		0.0470	0.0841	−0.0905	0.0480	0.0890	−0.0884	0.0475
#9		0.3598	1.6067	−1.0729	0.3307	1.7193	−0.8158	0.3453
#10		0.0074	0.0178	−0.0186	0.0072	0.0171	−0.0132	0.0073
#11		0.0413	0.0587	−0.0725	0.0412	0.0542	−0.0882	0.0412
#12		0.2716	0.9128	−0.7364	0.2432	0.9928	−0.8514	0.2574

**Table 5 materials-15-00880-t005:** Orthogonal analysis table of the magnetic flux density on the surface of fabrics cited.

Sample No.	Factor			Surface Magnetic Flux Density (mT)
	A	B	C	
#1	1	1	1	0.0031
#2	1	2	2	0.0562
#3	1	3	3	0.5571
#4	2	1	2	0.0041
#5	2	2	3	0.0590
#6	2	3	1	0.4152
#7	3	1	3	0.0063
#8	3	2	1	0.0475
#9	3	3	2	0.3453
#10	4	1	1	0.0073
#11	4	2	2	0.0412
#12	4	3	3	0.2574
Average Value K1j	0.2055	0.0052	0.1560	-
Average Value K2j	0.1594	0.0517	0.1327	-
Average Value K3j	0.1330	0.3892	0.1575	-
Average Value K4j	0.1020	-	-	-
Range R	0.1035	0.3841	0.0248	-
Optimal Level	A1	B3	C3	-

**Table 6 materials-15-00880-t006:** Magnetic flux density on the surface of fabrics after different washing times.

Washing Times	Magnetic Flux Density (mT)	Sample			
		#3	#6	#9	#12
0	Average value	0.5614	0.4169	0.3598	0.2716
	Maximum value of N pole	2.1781	1.3410	1.6067	0.9128
	Maximum value of S pole	−2.2112	−1.7747	−1.0729	−0.7364
5	Average value	0.5378	0.4033	0.3441	0.2633
	Maximum value of N pole	2.4340	1.0769	0.9781	0.8343
	Maximum value of S pole	−1.3266	−1.6129	−1.2187	−0.7068
10	Average value	0.5324	0.3920	0.3437	0.2604
	Maximum value of N pole	2.6576	1.2342	1.3602	0.9080
	Maximum value of S pole	−1.1857	−1.9018	−1.4524	−1.0514
15	Average value	0.5300	0.3845	0.3339	0.2490
	Maximum value of N pole	1.9414	1.2578	1.1014	0.7270
	Maximum value of S pole	−2.4779	−1.1470	−1.1090	−0.7630
20	Average value	0.5226	0.3806	0.3326	0.2452
	Maximum value of N pole	2.8603	1.4825	0.9391	0.6599
	Maximum value of S pole	−1.6647	−1.5848	−1.1512	−0.3864

## Data Availability

All data can be found within the manuscript.

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
