# Peer review of "Exploration of the Magnetic Flux Density on the Surface of Seamless Knitted Fabrics Manufactured with Magnetic Polypropylene Fibers"

_materials, 2022, doi:10.3390/ma15030880_

Round 1
Reviewer 1 Report
materials-1532759
The manuscript is devoted to studies on the magnetic flux density on the surface of knitted fabrics made of magnetic polypropylene fibers. The topic focussed on application of the studied materials in clothing with health care functions and application of magnetic health care textiles in medical therapies is interesting and worth publishing.
The manuscript is well written although the main part of the text consists of experimental set-up description with rather not extensive discussion. I think that it would be useful if the authors enlarge this section reaching beyond obvious conclusions e.g. the higher the content of magnetic powder in magnetic fibers, the better the magnetic properties of magnetic fibers, or the higher the content of magnetic fibers in magnetic fabrics, the higher the magnetic flux density of magnetic fabrics
In my opinion the influence of the type of stitch is quite interesting and worth more extensive discussion. It may be important not only from the effectiveness of the magnetic flux density, but also from the wearable functions of the material. Please add some more comments on this aspect, including discussion with regard to the literature data.
The model studies were carried out under constant humidity level (not given) and temperature of 20oC. Such parameters are not characteristic of human body. Please comment on the change of magnetic flux at temperature range 30-40oC.
Reviewer 2 Report
The Authors should read about how to prepare a research paper for a journal with such as high IF as Materials. The work entitled 'Exploration of the Magnetic Flux Density on the Surface of Seamless Knitted Fabrics Manufactured with Magnetic Polypropylene Fibers' is an engineering work, not a scientific one.
In the opinion of the reviewer, the presented work requires a thorough revision and consideration of scientific issues:
- Have the authors familiarized themselves with the WHO recommendations regarding to magnetic clothing?
- What was the reason for the use of graphene viscose?
- What kind of magnetic powder, what grain size and how was it introduced into the polypropylene yarn?
- Is this powder released from clothing, e.g. during the washing process? The answer to this question is important because the magnetic powders released from the clothes could damage washing devices (washing machines).
- According to the authors, what should be the optimal content of magnetic powder in the raw materials for the production of this type of clothing?
- How do the obtained results relate to the existing literature reports?
Reviewer 3 Report
Manuscript Exploration of the Magnetic Flux Density on the Surface of Seamless Knitted Fabrics Manufactured with Magnetic Polypropylene Fibers is very interesting for the readers, shows the novel approach toward current society health issues.
I recommend publication after minor revision.
I have the following comments:
- Due to the fact that the magnetic nanoparticles content in PP fibers has the main effect on the magnetic properties, you should inform readers what kind of nanoparticles you used (chemical structure, nanoparticles dimensions,…) and how the nanoparticles were applied in or on PP fibers.
- In the manuscript you mention that magnetic nanoparticles increase the temperature of fiber. Is this increase of temperature influenced by the increase of magnetic properties? Please add the mechanisms of temperature increase and combine both.
Page 1, row 6: according to the statistics…add reference
Page 2, row 2: what do you mean “in other countries?”
Round 2
Reviewer 1 Report
The manuscript may be accepted in the present form. I would only like to ask the authors for some small editing:
Please remove "foreign countries" (second line in page 2).
Please change the sentence below Figure 1 on page 3 for "The properties of knitted fabrics are in general related to their structure [12]. As can be seen from Figure 1, the stitches of 1+3 mock rib are flat and thick due to the floating..."
Please use the symbol of "magnetic flux density" in some places for better reading of the text. Some places contain far too many "magnetic" e.g. the first acapit in chapter 3.4.
Reviewer 2 Report
What did the authors want to explain by saying: 'As can be seen from Figure 1, the properties of knitted fabrics are related to the structure of knitted fabrics [12]',
There is still no explanation on how do the obtained results relate to the existing literature reports?
The authors wrote: In this paper, the relationship between surface magnetic flux density and stitch parameters of 12 seamless knitted fabrics was explored by orthogonal experiments and intuitive analysis, with specific results listed in Table 5.
It is very difficult to see what do the results are in Table 5, and it is also completely impossible to understand what the Authors illustrated in Figure 4
In the article we can find a sentences: ‘After magnetic nanoparticles were added to the polymer solution, magnetic fibers can be obtained through spinning. The magnetism of magnetic fabrics is influenced by the type of magnetic powder in magnetic fibers, the content of magnetic powder, and the content of magnetic fibers. In this paper, the magnetic fiber yarn was added with permanent magnetic nanoparticles. The magnetism of magnetic fabrics can be maintained persistently after the treatment of permanent magnetization’.
The description made by the Authors does not explain either what magnetic powder was used or in what amounts. It is a pity that the authors did not provide the correct characteristics of the yarn used in the tests.
In the summary, the Authors put forward very general engineering conclusions from the conducted research. In the opinion of the reviewer, the conclusion was reached only in point 4 of the summary (The reason is that the magnetic particle distribution in the magnetic fiber is uneven, the magnetic field on the fabric surface is regional, and the ambient temperature and humidity change affect the experimental data results), while in point 5 of the summary there is the missing depiction of the results presented in Table 6 and Figure 5.
I still believe that the article is a research report, with a lot of vague descriptions of little relevance and sometimes there is a lack of descriptions of the results of the research and their interpretation.
